# Comparison of Targeted (HPLC) and Nontargeted (GC-MS and NMR) Approaches for the Detection of Undeclared Addition of Protein Hydrolysates in Turkey Breast Muscle

**DOI:** 10.3390/foods9081084

**Published:** 2020-08-08

**Authors:** Liane Wagner, Manuela Peukert, Bertolt Kranz, Natalie Gerhardt, Sabine Andrée, Ulrich Busch, Dagmar Adeline Brüggemann

**Affiliations:** 1Department of Safety and Quality of Meat, Max Rubner-Institut, Federal Research Institute of Nutrition and Food, E.-C.-Baumann-Straße 20, 95326 Kulmbach, Germany; liane.wagner@mri.bund.de (L.W.); manuela.peukert@mri.bund.de (M.P.); bertolt.kranz@mri.bund.de (B.K.); sabine.andree@mri.bund.de (S.A.); 2Department of Food Analysis, Bavarian Health and Food Safety Authority (LGL), NMR, Veterinärstr 2, 85764 Oberschleißheim, Germany; natalie.gerhardt@lgl.bayern.de (N.G.); ulrich.busch@lgl.bayern.de (U.B.)

**Keywords:** ^1^H-NMR, GC-MS, HPLC-UV/VIS, metabolomics, food fraud, protein hydrolysate, free amino acid contents, ProHydrAdd

## Abstract

The adulteration of fresh turkey meat by the undeclared addition of protein hydrolysates is of interest for fraudsters due to the increase of the economic gain by substituting meat with low cost ingredients. The aim of this study was to compare the suitability of three different analytical techniques such as GC-MS and ^1^H-NMR with HPLC-UV/VIS as a targeted method, for the detection of with protein hydrolysates adulterated turkey meat. For this, turkey breast muscles were treated with different plant- (e.g., wheat) and animal-based (e.g., gelatin, casein) protein hydrolysates with different hydrolyzation degrees (15–53%: partial; 100%: total), which were produced by enzymatic and acidic hydrolysis. A water- and a nontreated sample (REF) served as controls. The data analyses revealed that the hydrolysate-treated samples had significantly higher levels of amino acids (e.g., leucine, phenylalanine, lysine) compared with REF observed with all three techniques concordantly. Furthermore, the nontargeted metabolic profiling (GC-MS and NMR) showed that sugars (glucose, maltose) and/or by-products (build and released during acidic hydrolyses, e.g., levulinic acid) could be used for the differentiation between control and hydrolysates (type, degrees). The combination of amino acid profiling and additional compounds gives stronger evidence for the detection and classification of adulteration in turkey breast meat.

## 1. Introduction

Meat is an important supplier of high-quality nutrients such as proteins, minerals and vitamins. Since it is sold on the market for a low price, which does not cover the increasing production cost, it is in focus for food fraud. Adulterators look for opportunities to increase the economic gain. Possibilities would be to misrepresent, use illegal supply chains and/or manipulate the food product, e.g., replace/substitute some, or all, premium quality materials with lower-grade, cheaper cuts of meat, meat from other species or nonmeat components (e.g., water, additives) [1,2,3]. The fraud can influence the consumers’ satisfaction and confidence (religious, moral, cultural), but worse it can be extremely dangerous for human health, e.g., causing illness, provoke allergies or even causing death (e.g., melamine scandal) [1,4]. Therefore, it is necessary to have reliable analytical methods to detect adulteration.

The water binding capacity of meat is strongly related to the rate of early postmortem metabolism and the ultimate pH value. It is lowest at a pH of 4.9–5.4, but will increase with increasing or decreasing of the pH [5]. The ultimate pH of poultry breast muscle is 5.67–5.69 and therefore, the water uptake capacity is high [6]. A simple exposure to just water can lead to weight gain of the meat and consequently increase the economic gain. For a fair market competition, the detection of such fraudulent practices is required. The traditional method to determine extraneous water in meat is to analyze the water/protein ratio [7]. In an untreated sample is the water/protein ratio of chicken as well as turkey breast muscle <3.40, of chicken legs between 4.05–4.30 and of turkey legs between 3.80–4.05 (Commission regulation No 543/2008) [8]. If water is added, the water/protein ratio will be higher.

For more than a decade an undeclared addition of protein hydrolysates to poultry meat or meat products could be observed. This way, the analytical protein content rises, masking the water addition [3]. Protein hydrolysates consist mainly of amino acids and possibly peptides. They are cheaper, better soluble and harder to detect than protein additions. Besides amino acids, hydrolysates contain additional compounds like carbohydrates, fatty acids and/or side products, which are formed during the hydrolyzation process. A suitable method is the detection of the free amino acid contents using high performance liquid chromatography with ultraviolet-visible detection (HPLC-UV/VIS). This technique is well established in laboratories and is used as an alternative to the official method in Germany (§64 LFGB: Determination of free amino acids in meat using gas chromatography with flame ionization detection (GC-FID)) [9].

Alternative analytical approaches such as nontargeted metabolomics provide an entire profile (chromatogram, spectrum, fingerprints, etc.) of a suspicious sample [10,11]. These methods have gained more and more in importance in recent years due to strong technical improvements. With these promising and valuable high-throughput tools such as mass spectrometry (MS) based techniques or nuclear magnetic resonance (NMR) spectroscopy, it is possible to identify and quantify small organic molecules with molecular weights of less than 1.5 kDa, including carbohydrates, peptides, nucleotides, lipids and amino acids [12]. In order to analyze such a broad spectrum of metabolites with diverse properties and concentrations (over several magnitudes) in just one single sample, it is important to have techniques, which are robust and sensitive. The major techniques are MS coupled to a chromatographic separation and NMR. The high sensitivity and selectivity make MS a powerful tool to detect molecular masses and fragmentation patterns for chemical structure identification. It is possible to profile myriads of metabolites due to the different combinations of separation, ionization and detection technique. On the other hand, NMR spectroscopy provides characteristic information on the metabolic profile by analyzing small amounts of one sample in a nondestructive, quantitative and short-time period way with convenient sample preparation [13]. However, independent of the used approach (targeted or nontargeted), large reference datasets are required to account for the natural variation in different products due to multiple influencing factors such as feeding or storage conditions (duration and temperature) [3,14].

Chemometrics is a powerful multivariate data analysis tool that reduces a huge amount of generated data by (1) grouping or ordering unknown samples with similar characteristics (qualitative) and (2) ascertaining adulterant analytes in sample (quantitative) or (3) for assessing their quality or authenticity [14]. Chemometrics, and for this purpose used clustering, principal component analyses (PCA) as well as regression analyses, is a routine complement for MS- and NMR-based metabolomics [15].

The objective of this study was to compare the suitability of three different analytical methods such as GC-MS and ^1^H-NMR (nontargeted approaches) and HPLC-UV/VIS as a targeted method, for the detection of adulterated turkey breast muscle with protein hydrolysates. HPLC-UV/VIS was selected due to the fact that it requires inexpensive equipment frequently available in every control laboratory. The first part focuses on the identification of specific markers such as amino acids, which are detectable with all three methods. The second part (nontargeted approaches) focuses on the identification of additional markers as well as possible classification of the added hydrolysates.

## 2. Materials and Methods

### 2.1. Chemicals

Sodium hydroxide, 5-sulfo salicylic acid, sodium chloride (NaCl, 99%), hydrochloric acid (HCl), sodium acetate, ninhydrin, hydrindantin-dihydrat, methanol and chloroform were obtained from Merck (Darmstadt, Germany). Ethylenediaminetetraacetic acid (EDTA, 99%) was purchased from AppliChem (Darmstadt, Germany) and tris(hydroxymethyl)aminomethane (TRIS Pufferan, Tris) from Carl Roth (Karlsruhe, Germany). The reagents (eluent buffers A, B, C, D, E, F; cleaning solution W, derivatization reagent R, sampling dilution buffer and autosampler solution) used for the amino acid (AA) determination were bought from membraPure (Henningsdorf, Germany). The AA standards (AA standards physiological: acidics–neutrals–basics), the amino acid mix solution (certified) and l-asparagine, l-glutamine, s-(2-aminoethyl)-l-cysteine hydrochloride (thialysine), l-norleucine, isopropanol, gelatin hydrolysate enzymatic, HyPep^®^ 4601 protein hydrolysate from wheat gluten, protein hydrolysate N-Z-amine^®^ AS, casein from bovine milk, gelatin from porcine skin and gluten from wheat were bought from Sigma-Aldrich (Steinheim, Germany). 2-Methoxyethanol was obtained from Riedel-deHaen (Seelze, Germany). For GC-MS derivatization methoxyamine hydrochloride (MAH) and pyridine were purchased from Sigma-Aldrich (Steinheim, Germany), and *N*-methyl-*N*-(trimethylsilyl)trifluoroacetamide plus 1% chlorotrimethylsilane (MSTFA + 1% TMCS) were purchased from Thermo Fisher (Dreieich, Germany). For NMR analyses reagents such as deuterium oxide (D_2_O) containing 0.05 wt% TSP (sodium-3-(trimethylsilyl)-2,2,3,3-tetradeuteriopropionate) and maleic acid were obtained from Sigma-Aldrich (Steinheim, Germany) and D_2_O (99.96%) was obtained from VWR (Ismaning, Germany).

### 2.2. Sampling and Adulteration of Turkey Breast

**Meat.** Three female turkeys (BUT Big 6, *Meleagris gallopavo*) with an average weight of 10.3 kg and in average 112 days old were collected at 4 °C from a slaughterhouse in Germany directly after slaughter. The *Musculus pectoralis superficialis* was taken after dissection [16]. After sampling, all meat pieces were immediately frozen in liquid nitrogen and stored at −20 °C until further use.

**Protein hydrolyzation.** The protein powders (1.0 g of casein from bovine milk, gelatin from porcine skin or gluten from wheat) were hydrolyzed at 150 °C in 8 mL of 6 M aqueous HCl solution for 1 h (total hydrolyzation: TH, degree of hydrolyzation: 100%). After that, neutralization with solid 1.0 M NaOH was performed. The final concentration of amino acids was 55.6 g/L (~0.5 M).

The protein hydrolysate powders (gelatin hydrolysate enzymatic, HyPep^®^ 4601 protein hydrolysate from wheat gluten, protein hydrolysate N-Z-amine^®^ AS (casein)) were used without further hydrolyzation (partial hydrolyzation: PH). All three bought protein hydrolysates were peptones (enzymatic hydrolysis with pepsin (E.C.3.4.4.1) or acidic hydrolysis). The degree of hydrolyzation was analyzed photometrically as the following describes. From each sample, 250 µL (aqueous AA solution) were added to 250 µL of 4 M acetate buffer (pH 5.5) and mixed. After that, 75 µL of 1 M NaOH and 250 µL ninhydrin solution (174 mg Ninhydrin + 28 mg hydrindantin-dihydrat in 10 mL 2-methoxyethanol) were added. The reaction took place at 95 °C for 20 min and was stopped by cooling the mixture in an ice bath (0 °C) for 20 min. 300 µL of each mixture was diluted with 1000 µL isopropanol solution (50% *v*/*v* in water) and measured at room temperature at 570 nm (photometer DU 640, Beckman Coulter GmbH, Krefeld, Germany). The degree of hydrolyzation (DH) was calculated by comparing the free amino groups of the samples with a solution of the same not hydrolyzed protein (0% DH) and total hydrolyzed protein (100% DH). That implies a hydrolyzation degree for gelatin, wheat and casein of 15% ± 3%, 16% ± 2% and 53% ± 2%, respectively.

**Hydrolysate injection.** About 1 mL solution (0.5 M) hydrolyzed protein or commercially available protein hydrolysate or water was injected per g turkey breast meat across and along the muscle fiber. The samples were frozen at −80 °C and lyophilized for GC-MS and ^1^H-NMR analyses. All samples were stored at −80 °C until further use.

**Sample code.** The sample codes were chosen as followed: reference sample without injection is called REF, an additional control sample injected with water is called water. The different protein hydrolysates (gelatin (G), wheat (W) and casein (C)) with different hydrolyzation degree (partial (PH) or total (TH)) are indicated with the following codes: GPH, WPH and CPH for partial, respectively, and GTH, WTH and CTH for total hydrolyzation, respectively. This means, for example, that the sample GPH contained protein hydrolysate gelatin and was partially hydrolyzed.

### 2.3. Amino Acid Analysis Using HPLC-UV/VIS

#### 2.3.1. Sample Preparation for Amino Acid Analysis

The frozen turkey samples (2 g each) were homogenized with an Ultra-Turrax T10 (12,000 rpm, IKA Werke GmbH und CO KG, Staufen, Germany) in 5 mL 0.025 M EDTA/0.100 M Tris buffer (pH 8.0) at −20 °C. As internal standards, l-norleucine and l-thialysine were used with a final concentration of 133 µM and 88 µM, respectively. The proteins and longer peptides were precipitated with 30% *v*/*v* of 15% 5-sulfosalicylic acid over 30 min at 4 °C at a resulting pH of about 2.2 (pH less than 2.5 is recommended for cation exchange chromatography). The samples were centrifuged (15 min, 4 °C, 6827× *g*), filtered (45 µm) and then frozen at −20 °C until further use. The centrifugation and filtration were repeated directly before the analysis.

#### 2.3.2. Amino Acid Content Determination

The samples were analyzed in duplicates with internal (l-norleucine- and l-thialysine-solution, see Section 2.3.1) and external AA standards. The external standards were worked up similar to the meat samples. A regularly calibration of the amino acid analyzer was done with a certified AA standard (amino acid mix solution [17]). For the cation exchange chromatography, a column with 3 µm beads with a separation over a pH-range from 2.9 to 10.4 and ninhydrin post column derivatization was performed. An injection volume of 20 µL (pH 2.2) and a flow rate of 180 µL/min were applied. For the spectrophotometric analysis two photometers with wavelengths of 440 nm and 570 nm were used for detection of the free amino acids. The limit of detection (LOD) and quantification (LOQ) were defined as three and ten times the signal to noise ratio of the external standard solution, respectively. All measured contents of the free amino acids (FAA) were above the LOQ (0.13 mg/100 g–0.33 mg/100 g, depending on the different AA).

### 2.4. GC-MS-Based Metabolomics Analyses

#### 2.4.1. Sample Preparation for Metabolomics Study of White Breast Meat

The sample set consisted of study samples, mix samples and blanks. The mix samples were prepared by combining an aliquot from each muscle powder and served for normalization. All sample types were extracted in the same manner. Study samples were extracted in triplicates. In a first step 20 mg of the dried, homogenous meat powder was extracted with 600 µL ice-cold 80% methanol containing 10 internal standards using a bead mill homogenizer (Minilys, Bertin Technologies SAS, Montigny-le-Bretonneux, France) for 2 times 30 s and an ultrasonic bath for 2 min. The raw extract was centrifuged at 15,000× *g* for 20 min at 4 °C. In a second step the pellet was re-extracted with 600 µL ice-cold methanol:chloroform (2:1 *v*/*v*) according to the first extraction step. Both supernatants were combined, mixed and centrifuged at 15,000× *g* for 20 min at 4 °C. 100 µL of the supernatant were transferred into 2 mL glass vials containing a 200 µL glass insert and evaporated in a vacuum centrifuge (Christ Speedvac RVC 2¨C18 CD plus, Germany). The dried samples were stored under protective argon atmosphere at −80 °C until analysis.

#### 2.4.2. GC-MS Measurements and Data Processing

Prior to measurement samples were derivatized by methoximation and trimethylsilylation. For methoximation samples were shaken in 30 µL of a 20 mg/mL solution of MAH in pyridine at 50 °C for 1 h. Subsequently, 70 µL MSTFA+1%TMCS were added and samples shook at 70 °C for 1 h. GC-MS analysis was performed on a Shimdazu GCMS QP2010 instrument (Shimadzu, Duisburg, Germany) equipped with an OPTIC-4 injector (GL Sciences, Eindhoven, The Netherlands). A 1 µL aliquot of each sample was injected in a 1:7 split ratio. A series of n-alkanes (C7-C30) was used as a retention time standard. Analytes were separated on a 30 m Rxi-5SIL MS column containing a 10 m Integra-Guard column (Restek, 0.25.mm i.d., 0.25 µm film thickness), and with a linear temperature gradient starting from 80 °C to 300 °C with 5 °C/min and a final 5 min hold at 320 °C. Masses between 60 and 600 *m*/*z* were scanned.

For data analysis only those features were selected that were not present in blank samples. The annotation of compounds was performed using the NIST 14 library database implemented in GCMSsolution (Shimadzu, Duisburg, Germany). The retention times and selected masses used for relative quantification are listed in Appendix A. Samples were normalized according to the calculated means of the mix samples for each feature to reduce the impact of device maintenance (instrument tuning, liner exchange and septum exchange during the measurement batch). Briefly, signal intensities of the mix-samples within one measurement period (between device maintenance) were averaged as well as for the whole measurement batch. Using these means a correction factor was determined between the total means and the means for each measurement period between device maintenances. These correction factors were then applied to the corresponding features of study samples and calibration samples.

#### 2.4.3. Amino Acid Quantification

For AA quantification a calibration curve consisting of a reference standard mixture and 8 final concentrations in the range 1.96–250 pmol/µL was applied. These standard mixes were spiked into aliquots from pooled REF samples (50 µL of standard mixture into 50 µL of REF mixture) to reduce for the impact of the sample matrix. The calibration samples were prepared in duplicates. Calibration samples were evaporated in a vacuum centrifuge and prepared for GC-MS measurement according to Section 2.4.2. For quantification, the calibration samples were at first normalized and then the mean values (ion counts) of selected quantitative ions from calibration-reference samples were subtracted from calibration samples containing the standard mixtures. Calibration coefficients were R^2^ > 0.99 with acceptance of alanine which had an R^2^ = 0.97). The retention times and selected masses used for quantification are listed in Appendix A.

### 2.5. ^1^H NMR-Based Metabolomics Analyses

#### 2.5.1. Sample Preparation for Metabolomics Study of White Breast Meat

The same dried, homogenous breast muscle samples as used for GC-MS were prepared for NMR metabolomics as described previously by Wagner et al. [18] with slight modifications. In brief, 20 mg of lyophilized, grinded, homogeneous muscle powder was extracted using first ice-cold methanol, then ice-cold chloroform and finally ice-cold water (400 µL of each solvent). The samples were vortexed for 1 min and stored on ice for 10 min between each step, and then stored at 4 °C overnight for separation and finally after centrifugation (2000× *g*, 4 °C, 30 min) the aqueous phase was collected in a new tube. The collected samples (750 µL) were dried using a vacuum centrifuge. The dried samples were redissolved with 550 µL D_2_O, 25 µL MilliQ water and 25 µL D_2_O containing 0.05 wt% TSP as internal standard for quantification and chemical shift reference.

#### 2.5.2. ^1^H NMR Spectroscopy, Data Processing and Identification of the Signals

All samples were analyzed with a Bruker 400 MHz spectrometer (Bruker BioSpin GmbH, Rheinstetten, Germany). For the aqueous white breast muscle, a noesygppr1d pulse program at 25 °C with 64 scans, a spectral width of 8224 Hz collected into 65,536 data point and acquisition time of 3.98 s and an interscan relaxation delay of 4 s was used. ^1^H-^1^H COSY, ^1^H-^1^H TOCSY and ^1^H-^13^C HSQC were obtained on one representative muscle sample for metabolite identification purposes.

All data were processed using Bruker Topspin 3.6.0 software (Bruker), Fourier-transformed after multiplication by line broadening of 0.30 Hz and subsequently referenced to standard peak TSP at 0.00 ppm. After spectral phase and baseline were corrected, each NMR spectrum was integrated using Matlab R2017b (Mathworks, Natick, MA, USA) into 0.01 ppm integral regions (buckets) between 8.60 ppm and 0.80 ppm (area between 4.75 ppm and 4.80 ppm corresponding to the water signal was excluded). Each muscle spectrum region was scaled to the intensity of internal standard (TSP) for quantitative measurements. Afterwards, the signals were identified using ChenomX NMR Suite 8.4 library (ChenomX Inc., Edmonton, AB, Canada), the Human Metabolome Database (www.hmdb.ca) and previous literature [18,19,20] and confirmed with 2D-NMR in case of multiplicity. For quantification (profiling approach), 86 metabolites were identified by overlapping with standard spectra (Appendix A) and their concentrations (µmol/mg) were calculated using ChenomX NMR Suite 8.4 library after accounting for overlapping signals. The absolute concentrations were presented as mg/100 g wet weight (Appendix A).

### 2.6. Statistical Analysis

Multivariate data analyses were performed for the GC-MS data, the NMR spectral data (buckets) and the absolute concentrations of the metabolites (profiling approach) using the Simca-P+ software (version 13.0; Umetrics, Umeå, Sweden). All variables were centered and “pareto-scaled” (Par) (GC-MS data and NMR spectral data) or “unit variance” (UV)-scaled (NMR data, absolute concentrations). Principal component analysis (PCA) was used to screen the data and search for outliers. Outliers were determined using PCA-Hotelling T^2^ Ellipse (95% confidential interval (CI)).

All statistical calculations such as one-way analysis of variance (ANOVA) and Dunnett’s test were done in JMP (13.1.0, SAS Institute Inc., Cary, NC, USA).

The amino acid contents (alanine, leucine, methionine, phenylalanine, proline, serine, tyrosine, histidine, lysine and glutamate) determined by HPLC, GC-MS and NMR are presented as mg/100 g wet weight, henceforth referred to as mg/100 g. All data presented are mean ± standard deviation and differences were considered significant when *p* < 0.01.

## 3. Results and Discussion

The aim of this study was to compare different analytical techniques in respect to their performance for the detection of undeclared protein hydrolysates in fresh turkey breast. For this purpose, a traditional HPLC-UV/VIS approach focusing on the detection of free proteinogenic amino acids was compared with two nontargeted metabolic profiling techniques, GC-MS and ^1^H-NMR. Additionally, both nontargeted approaches were compared for their suitability in the detection of the adulterated turkey breast muscle with protein hydrolysates.

### 3.1. Capability of Amino Acid Profiling for the Detection of Added Protein Hydrolysates

The contents of the ten free amino acids (FAA) alanine, leucine, methionine, phenylalanine, proline, serine, tyrosine, histidine, lysine and glutamate out of the 20 proteogenic AA were analyzed by all methods. Those ten AA were selected as exemplary free amino acids with different properties, e.g., aliphatic (alanine, leucine, proline), aromatic (phenylalanine, tyrosine), acidic (glutamate), basic (lysine, histidine), hydroxylic (serine) and sulfur-containing (methionine). The results of five groups (REF and addition of water, gelatin-, wheat- and casein-hydrolysates) were compared depending on the hydrolyzation degree (PH or TH).

The contents of FAA for the addition of partial and total protein hydrolysates are shown in Figure 1 and Figure 2, respectively. All these data are also summarized in the Appendix A. The reference samples (REF) were not modified and variations were therefore only occurring through the analytical error and natural variations of the FAA contents. The natural FAA contents depend on several conditions, e.g., the gender of the birds [21] or special feed additives [22].

It is obvious that the amount of FAA contents determined by HPLC-UV/VIS (in mg/100 g) and GC-MS as well as ^1^H-NMR (in mg/100 g) are quite different. The FAA contents determined by GC-MS and ^1^H-NMR are on average 3.8 times (range: 0.6-fold to 7.0-fold) and 3.3 times (range: 0.1-fold to 10.2-fold) higher, respectively, compared with the contents determined by HPLC-UV/VIS. The differences can be arising from the different sample preparations. One possibility could be that the homogenization method (HPLC-UV/VIS) was not able to dissolve all FAA. The homogenization method for HPLC did not contain any specific extraction step with solvents, whereas for GC-MS and ^1^H-NMR the samples were extracted using mixtures of water, methanol and chloroform. Moreover, it is possible that the extraction method (GC-MS and ^1^H-NMR) led to protein hydrolysis. However, the different quantification procedures could also have caused these differences. Comparative experiments (e.g., with protease inhibitors) clearly demonstrated that no protein hydrolysis occurred by using the homogenization method (manuscript in preparation). This method was also used to determinate 18 of the 20 proteinogenic FAA contents of chicken breast meat (manuscript in preparation) and the contents were in agreement with Rikimaru and Takahashi [23].

The addition of water to the turkey breast muscle resulted in tendentially lower contents of FAA, but statistical significance was not reached by using the Dunnett’s test (Figure 1 and Figure 2 and Appendix A). The reduced mean values in water treated samples could be explained as dilution or even wash-out effect. When the amount of water injected to the sample exceeds its water binding capacity, some endogenous compounds (e.g., FAA) might be washed out.

#### 3.1.1. Comparison of Partial Hydrolyzed Wheat-, Gelatin- and Casein-Hydrolysates

Partial enzymatic hydrolysates from gelatin (GPH, hydrolyzation degree 15% ± 3%), wheat (WPH, hydrolyzation degree: 16% ± 2%) and casein (CPH, hydrolyzation degree 53% ± 2%) were added to the meat samples, respectively (Figure 1). WPH and GPH were only slightly hydrolyzed and therefore most of the protein was converted to peptides. Therefore, the amount of FAA in these two hydrolysates was lower compared to CPH, which was hydrolyzed to a higher hydrolyzation degree.

Hence, nearly no significant differences were found for the FAA contents of GPH and WPH related to the REF. Only the content of free lysine (^1^H-NMR, from 5.26 mg/100 g ± 0.45 mg/100 g (REF) to 26.9 mg/100 g ± 3.37 mg/100 g) for GPH, as well as the free methionine (HPLC-UV/VIS) and free leucine contents (^1^H-NMR) for WPH showed significant differences (Appendix A).

Contrary to this, the CPH showed clearly significant different FAA contents compared to the REF. As determined with HPLC-UV/VIS-method, the FAA contents of leucine, methionine, phenylalanine and histidine were highly significant different (*p* < 0.001) and for serine significant different (*p* < 0.01). For example, the FFA content increased for leucine from 1.97 mg/100 g ± 0.33 mg/100 g (REF) to 19.9 mg/100 g ± 2.39 mg/100 g (CPH). The FAA contents analyzed with GC-MS showed significant differences for five of the ten listed FAA. Leucine was also highly increased (6.97 mg/100 g ± 2.78 mg/100 g for REF to 111 mg/100 g ± 26.4 mg/100 g (CPH), *p* < 0.001), as well as methionine, phenylalanine and lysine. The histidine content was increased significantly (*p* < 0.01). The analysis with ^1^H-NMR showed seven increased FAA contents: leucine, methionine, phenylalanine, proline and lysine increased highly (*p* < 0.001), whereas alanine and serine increased significantly (*p* < 0.01). For example, leucine increased from 4.26 mg/100 g ± 1.25 mg/100 g (REF) to 78.6 mg/100 g ± 19.6 mg/100 g (CPH).

For a clear proof, several FAA contents should differ significantly from the reference sample. In this way other reasons for different FAA contents (e.g., feed supplementation with AA) can be excluded with higher probability. It can be concluded in this study, that in case of partial hydrolysate treatment the detection of fraud could not be ensured by using only these ten FAA contents.

#### 3.1.2. Comparison of Total Hydrolyzed Wheat-, Gelatin- and Casein-Hydrolysates

All the total hydrolyzed proteins (GTH, WTH, CTH) showed a hydrolyzation degree of 100% and were therefore only composed of AA. As expected, the FAA contents of all hydrolysate-treated samples were increased dramatically compared to the REF (Figure 2). The analysis with HPLC-UV/VIS revealed two significant and seven highly significant increases of FAA contents for GTH, eight highly significant increased FAA contents for WTH and nine rises of the amounts of FAA for CTH (*p* < 0.001). For example, the content of proline changed from 2.56 mg/100 g ± 0.42 mg/100 g (REF) to 52.5 mg/100 g ± 20.4 mg/100 g (GTH), 44.9 mg/100 g ± 16.8 mg/100 g (WTH) and 39.2 mg/100 g ± 13.8 mg/100 g (CTH), respectively. With GC-MS, only highly significant changes were found: for GTH seven, for WTH nine and for CTH all ten FAA contents were increased. As for HPLC-UV/VIS, the proline content was raised obviously: from 10.0 mg/100 g ± 5.81 mg/100 g to 134 mg/100 g ± 71.4 mg/100 g (GTH), 161 mg/100 g ± 31.8 mg/100 g (WTH) and 160 mg/100 g ± 30.5 mg/100 g (CTH), respectively. The same results were also found for the ^1^H-NMR analysis: one significant and eight highly significant increases for GTH (proline: from 5.06 mg/100 g ± 0.78 mg/100 g (REF) to 77.5 mg/100 g ± 16.3 mg/100 g), and nine highly significant differences for WTH (proline: 72.8 mg/100 g ± 12.1 mg/100 g) and CTH (proline: 53.8 mg/100 g ± 16.1 mg/100 g). Therefore, only tyrosine (GTH), alanine and lysine (WTH) as well as alanine (CTH) showed no significant differences determined by HPLC-UV/VIS. For the GC-MS-method, only methionine, tyrosine, histidine (GTH) and lysine (WTH) were not significantly different. The analysis by ^1^H-NMR revealed also nearly exclusive significant differences with only few exceptions (tyrosine, histidine for GTH, histidine for WTH and CTH).

Depending on the hydrolysate type, different AA were more affected. The addition of GTH resulted in higher levels of alanine, whereas WTH showed higher levels of glutamate and CTH higher levels of leucine, methionine, tyrosine and lysine (Figure 2 and Appendix A). The latter AA might be used as an indicator for animal-based protein origins whereas glutamate could indicate plant-based protein origins.

It was shown in this study that in case of total hydrolysate treatment a general detection of fraud is possible.

#### 3.1.3. General Aspects Regarding the Detection of Free Amino Acids in Treated Breast Muscles

All three methods used (HPLC-UV/VIS, GC-MS, ^1^H-NMR) showed comparable results. Although the FAA contents of the first method were about three- to fourfold lower compared to the other two methods, the validity was given. This is due to the fact that all samples were compared to the corresponding REF, determined with the same method. Hence, the method of sample preparation plays an important role for absolute quantities, whereas regarding the differentiation between REF and hydrolyzed-treated samples (rations are kept independently of the sample preparation) the method has no impact.

It can be concluded that the differentiation of hydrolysate addition depends on the degree of hydrolyzation. If breast muscles were treated with low degree hydrolysates, the additional injected FAA might not induce a significant increase over the range of natural variation. It was shown that a high hydrolyzation degree significantly increased the free AA content of several AA independently of which analytical technique (HPLC-UV/VIS, GC-MS or ^1^H-NMR) was used.

Specific FAA profiles might be used for a tentative classification of the origin of the hydrolysate type (e.g., plant-based vs. animal-based protein origins). Nevertheless, a clear classification and identification of the protein used for hydrolyzation was not possible. Thus, it is of interest whether further information about additional compounds might be helpful for the detection and classification of the hydrolysates. For this, the following hypotheses for section two were postulated: (1) Original protein sources are not clean and contain additional compounds, which can be introduced into the breast meat. (2) Acidic hydrolysis leads to formation of byproducts and these compounds are also possible to be found in the breast meat. (3) Additional metabolites can be washed out from the breast meat and (4) therefore, information from metabolite profiling might be of interest and was included in the analysis.

### 3.2. Metabolomics Approaches to Obtain Additional Information Regarding Hydrolysate-Treated Samples Independently of the Hydrolyzation Degree

The detection of hydrolysate treatment in turkey breast muscle by amino acid profiling largely depends on the hydrolyzation degree. Our results clearly indicated that an addition of total hydrolyzation increases the free amino acid content tremendously (Figure 2) so that a detection with all three presented methods was possible. However, the lower the hydrolyzation degree the more uncertain is the validity of amino acid profiles between the natural variation and the differentiation due to hydrolysate treatment. Therefore, we applied two nontargeted metabolite profiling approaches (GC-MS and ^1^H-NMR) to test for their suitability in the detection of hydrolysate treatment in turkey breast muscle. Both approaches allow to detect additionally several metabolites besides the proteogenic amino acids like carbohydrates, organic acids, lipids, et cetera. For both techniques PCA was used to check for the variation of metabolite profiles between controls and treatments and between the different types of hydrolysates (Figure 3).

With GC-MS a total of 129 features were considered for PCA. The first (horizontal) and second (vertical) components explained 38.0% and 15.7% of the variation, respectively, with R^2^X = 98.4%, Q^2^ = 72.8% (Figure 3a). A clear separation between the controls (REF and water-treated control) and five of the hydrolysate-treated sample groups was observed. GPH did not segregate from the controls. WPH varied only in PC2 direction whereas CPH, with 53% hydrolyzation degree, stronger differentiated in PC1 direction. The total hydrolysate treated sample groups showed most variation in PC1 direction.

In addition, the ^1^H-NMR spectra obtained were compared by PCA (PC1 vs. PC2) (Figure 3b). The first component (horizontal) which is explained by 40.8% of spectral variation clearly separates controls (REF and water-treated control, left) with total hydrolyzed-treated samples (right) and partial hydrolyzed-treated samples (middle). The second component explained 21.9% of variation and separates the controls and total hydrolyzed-treated samples (top) from partial hydrolyzed (bottom) samples. The model parameters were the following: R^2^X = 98.5%, Q^2^ = 93.4%, 16 components. In order to identify metabolic changes, the absolute concentrations of 86 metabolites were quantified through a profiling approach from ^1^H-NMR spectra.

The total hydrolysate treated samples clearly separated from the controls in PC1 observed with both techniques. Interestingly, with GC-MS analysis the wheat and casein origins showed higher similarities to each other compared to gelatin (GTH). Whereas for NMR analyses, higher similarities were observed between GTH and WTH. Obviously, independently of the analytical technique, there is a clear separation between gelatin (GTH) and casein (CTH). From the loading plot (Figure 3c,d), it can be deduced that proteinogenic amino acids particularly contribute to the differentiation of the total hydrolysate treated samples and the controls (in PC1), which is in accordance to the results presented in Section 3.1. Nevertheless, besides the amino acids other compounds could be identified which play an additional role for the variation in PC1 such as hydroxyproline, levulinic acid, ornithine or glycerol (the complete feature tables are presented in Appendix A). Compounds, which were additionally detected by ^1^H-NMR were pyruvate and acetate. Further compounds detected by GC-MS were 5-hydroxylysine, 3-MCPD or aminomalonic acid among several nonidentified molecular features. These additional compounds represent characteristics of the protein origin or are byproducts formed during the acidic hydrolyzation process. In addition to the amino acid profiles these byproducts might contribute to a better classification of the protein sources.

The wheat protein source contained higher amounts of sugars, which was also observed for the WPH treated breast samples (see below). During the acidic hydrolysis of the protein source, the sugars contained therein such as maltose, saccharose, glucose or fructose are converted to levulinic acid in presence of hydrochloride and under high temperature [24,25]. Thus, the high levels of levulinic acid detected in our analyses could be used to differentiate plant-based hydrolysates and as a marker for acidic hydrolyzation treatment. Nevertheless, other plant-based hydrolysates need to be tested for their carbohydrate content in contrast to animal-based protein sources.

The total protein hydrolysate from gelatin contained higher amounts of AA derivates such as hydroxyproline and hydroxylysine. Gelatin is the denatured form of collagen, which is one of the most abundant proteins in meat, ranging between 2 and 4 mg/g in chicken breast meat [26]. Most abundant amino acids of collagen are glycine, proline, glutamate and hydroxyproline [27]. Hydroxyproline is specific to collagen and its concentration in collagen is rather constant with ~12% [28]. Therefore, hydroxyproline is used to estimate the connective tissue content [29,30]. Regarding the treatment of turkey breast meat with protein hydrolysates, the hydroxyproline content can serve as a marker for animal-based protein sources such as gelatin. Hydroxylysine is another modified amino acid, which is unique to collagens. Similar to hydroxyproline, this amino acid becomes posttranslational hydroxylated and subsequently glycosylated forming the α-helical structure of collagens [31]. Therefore, 5-hydroxylysine might serve as an additional indicator of gelatin hydrolysate treatment. Aminomalonic acid that was most abundant in GTH followed by CTH treated samples represents an amino acid derivative, whose origin is suspected to be related to protein oxidation processes [32] and to play a role in the serine-glycine interconversion [33]. According to our results, the acidic hydrolysis process might increase the formation of aminomalonic acid in dependence of the protein source, namely the glycine-rich gelatin.

For casein, the second tested animal derived hydrolysate, it was not that a particular molecule was strongly increased, but the combination of several molecular features could hint towards this treatment. In addition to the amino acid profile, the casein treated samples had higher levels of 3-MCPD and a number of not-identified molecular features (Appendix A).

Interestingly, the acidic treatment of the different protein sources led to the formation of 3-MCPD. This compound can be found in numerous foodstuffs and is described to be present in acidic hydrolysates of proteins [34]. Depending on the remaining lipids in the original protein sources, different amounts of 3-MCPD and 3-MCPD fatty acid esters might be formed and injected into the breast meat. Thus, 3-MCPD represents an additional marker for acidic hydrolysis, similar to levulinic acid.

From the score plots of the partial hydrolysate treated breast muscles, we observed a clear separation of all three sample groups using ^1^H-NMR technique, whereas by GC-MS analysis the GPH group largely overlapped with the controls. The WPH treated samples were in a medium distance to the controls, and the highest variation to the controls was observed for CPH treated samples. Those observations are in accordance with the different degrees of hydrolyzation in the partial hydrolysates with GPH having a hydrolyzation degree of 15% and CPH having a hydrolyzation degree of 53%. Even though WPH has a hydrolyzation degree of only 16%, the better separation compared to GPH might be explained by the plant-based origin and the present additional metabolites.

A closer look at the loading plots from the PCA models for control samples and partial hydrolysate treated samples indicated that proteinogenic amino acids play a minor role for the variation between controls (and GPH) and WPH, both having a low hydrolyzation degree (15% and 16%, respectively). The variation of CPH from controls, which was to 53% hydrolyzed, was already dominated by proteinogenic amino acids. Especially the plant-based hydrolysate contained additional sugars such as maltose (Figure 4) and hexoses like glucose, detected with both technical approaches. In addition, higher levels of glycerol were detected in WPH. As mentioned above, future studies will have to elucidate to which extent different sugars are present in plant-based protein extracts used for hydrolysis. With the animal-based protein hydrolysates, GPH and CPH, the contents of ornithine (Figure 4) were increased as detected with ^1^H-NMR and GC-MS. A low level of levulinic acid and 3-MCPD was observed in CPH treated breast muscle, which might be related to the CPH production process (CPH was commercially obtained). Using ^1^H-NMR technique, higher amounts of acetate (GPH, WPH, CPH), butyrate (CPH), carnitine (WPH, CPH), citrate (WPH, CPH), glutathione (WPH, CPH), pantothenate (CPH) and putrescine (GPH, WPH, CPH), myo-inositol (GPH, WPH) were additionally detected (Figure 4 and Appendix A), whereas with GC-MS analysis we obtained increased levels of oxoproline (CPH), urea (CPH), hydroxylysine (GPH) and malic acid (GPH, WPH, CPH) among a few nonidentified compounds (Figure 4 and Appendix A). It can be concluded that the lower the hydrolyzation degree the more important are the additional compounds from the protein origins for the differentiation of nontreated samples and hydrolysate treated samples.

In accordance with the reduced AA content in water treated samples, we observed for several endogenous metabolites of turkey breast muscle a similar reduction when samples were injected with the different kinds of hydrolysates. This effect was particularly obvious for highly water-soluble compounds such as creatinine and lactate, which were detected by GC-MS and ^1^H-NMR. Additionally, reduced levels (not significant) of myo-inositol and the peptides glutathione and anserine were detected by ^1^H-NMR. With GC-MS profiling we detected reduced levels of 4-hydroxybutanoic acid, myo-inositol, inosine and uracil among several nonidentified molecular features (Appendix A). In our approach, sample preparation was performed using ~2 g fresh turkey breast meat to minimize the effect of natural variation when comparing different hydrolysate types. Whether the observed wash-out effect can also be detected by using whole breast muscles has to be validated by further studies. It can be suspected that the natural variation has a greater impact than the detected small levels of a wash-out.

## 4. Conclusions

This study aimed at a comparison between different analytical methods and their possibility to detect adulteration of turkey breast meat with different hydrolysates. It showed that FAA profiling allows for the detection of protein hydrolysate treatments only above a certain threshold which is mainly related to the degree of hydrolyzation. The samples naturally strongly differ in their free amino acid contents as a result of feeding, genotype and meat age. Therefore, the FAA analyses under these conditions (e.g., determination of only ten FAA contents) were not suitable for the detection of food fraud in the case of partial hydrolysates. To overcome this limitation, the contents of more than ten FAA of the 20 proteinogenic AA should be analyzed. Further on, a much higher quantity of samples ought to be measured. The evaluation of these datasets enables the reduction of the variations and therefore more significant differences. The additions of hydrolysates with high amounts of AA to breast meat were easily proofed within this study.

The different profiling techniques revealed that protein sources contain different metabolites, which can be used as biomarkers for the detection of partial hydrolysates. Furthermore, byproducts formed during acidic hydrolysis provide additional evidence for the treatment of breast meat with protein hydrolysates. Therefore, a combination of FAA and metabolite (by-products) profiling makes it possible to identify and classify the addition of nondeclared hydrolysates to turkey breast meat. In addition, an advantage of this comprehensive analysis is that it might be possible to proof the addition of animal-based proteins or animal-based hydrolysates to vegetarian or vegan products. According to the advantages of NMR in terms of sample throughput and direct quantification of the identified compounds, ^1^H-NMR is used in further/detailed studies analyzing food fraud of turkey breast meat.

## Figures and Tables

**Figure 1 foods-09-01084-f001:**
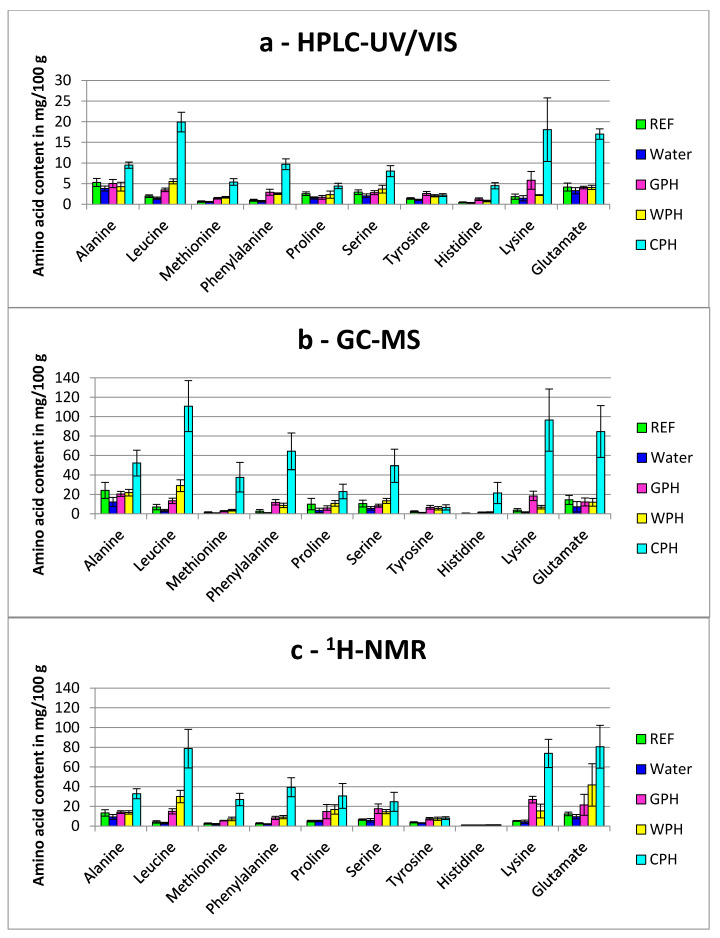
Free amino acids contents (mean ± standard deviation) of turkey breast meat samples treated with and without the addition of partial protein hydrolysates or water and analyzed via: (**a**) HPLC-UV/VIS (**b**) GC-MS (**c**) ^1^H-NMR. Sample codes: REF: Reference, Water: injected with water, GPH: partial hydrolysate gelatin, WPH: partial hydrolysate wheat; CPH: partial hydrolysate casein.

**Figure 2 foods-09-01084-f002:**
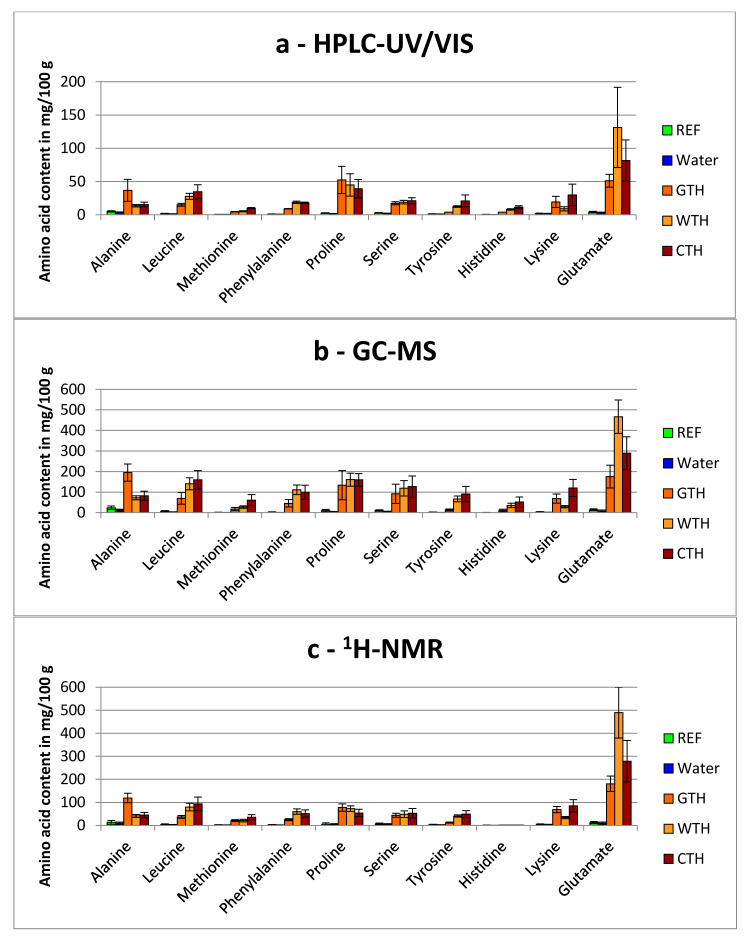
Free amino acids contents (mean ± standard deviation) of turkey breast meat samples treated with and without the addition of total protein hydrolysates or water and analyzed via: (**a**) HPLC-UV/VIS (**b**) GC-MS (**c**) ^1^H-NMR. Sample codes: REF: Reference, Water: injected with water, GTH: total hydrolysate gelatin, WTH: total hydrolysate wheat; CTH: total hydrolysate casein.

**Figure 3 foods-09-01084-f003:**
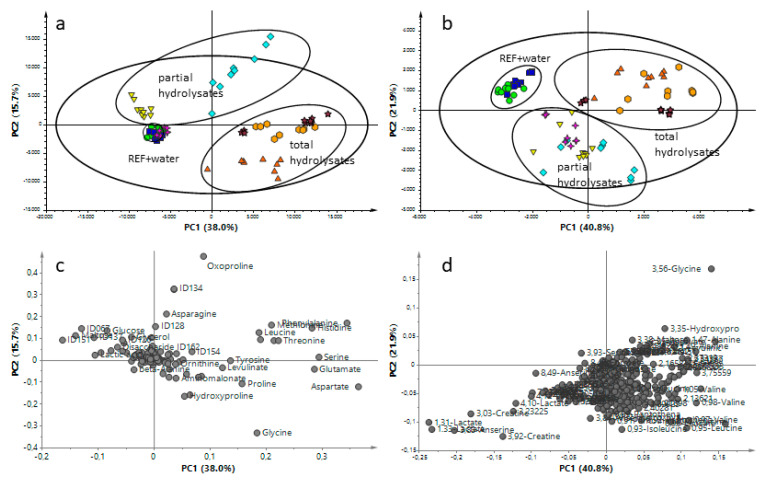
The score (**a**,**b**) and the loading (**c**,**d**) plots of all data of the principal component analysis (PCA) based on the metabolic fingerprint from adulated turkey breast meat with different protein hydrolysates (type and degrees) analyzed by GC-MS (**a**,**c**) and ^1^H-NMR (**b**,**d**). REF (
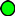
), Water (
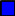
), GPH (
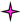
), WPH (
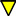
), CPH (
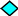
), GTH (
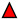
), WTH (
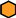
) and CTH (
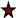
). GC-MS: The first component is explained by 38.0% and the second component by 15.7% of the variation (model parameters: R^2^X = 98.4%, Q^2^ = 72.8%, 17 components). ^1^H-NMR: The first and second components explained 40.8% and 21.9% of variation, respectively (model parameters: R^2^X = 98.5%, Q^2^ = 93.4%, 16 components).

**Figure 4 foods-09-01084-f004:**
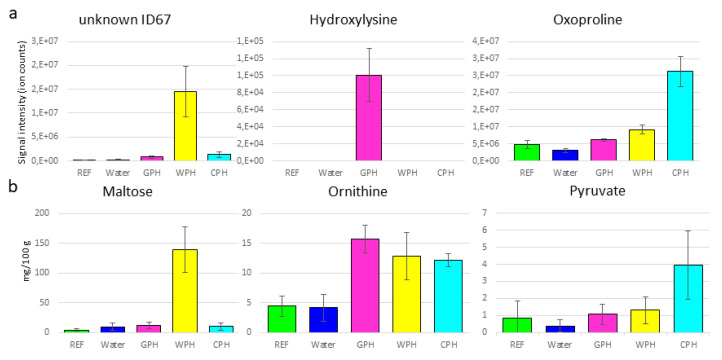
Selected signals that were present in partial hydrolysates and show significant differences between references and hydrolysate treated samples obtained via GC-MS (**a**) and ^1^H-NMR (**b**).

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
