# Peer review of "Comparison of Targeted (HPLC) and Nontargeted (GC-MS and NMR) Approaches for the Detection of Undeclared Addition of Protein Hydrolysates in Turkey Breast Muscle"

_foods, 2020, doi:10.3390/foods9081084_

Round 1

Reviewer 1 Report

Adulteration of food is one of the main elements of the economic struggle.

The producers do not care about the nutritional value or biologically active compounds which join in food.

The most important element is the level of protein and energy in the next stage, other elements such as fat content of ash or other compounds, e.g. allergens. Analytical methods are not sufficient because using reference methods we generally analyze the content of nitrogen or fat. We do not receive information from what source they come from.

The use of labeled antibodies is very expensive and only confirms the adulteration. The use of chromatography requires additional equipment, often extraction takes more time than analysis alone. There is interference and falsification of the result. Therefore, we are looking for accurate qualitative and quantitative methods.

Unfortunately, sometimes this is not possible to do. The use of molecular techniques is sometimes impossible and very expensive. The introduction of techniques based on spectrum or image analysis definitely accelerate the analytical process

In addition to magnetic resonance imaging, near infrared methods are also used. The reviewed publication is a good scientific study. It brings us closer to the application of new research techniques. The strength of the work is the analytical description of the methods used

The only problematic element is the cost of purchasing equipment and its application in routine tests. Today it seems we can apply these methods in reference laboratories, only.

In conclusion, this publication has research elements, introduces new elements in the development of research on nutrition.

Author Response

Dear Reviewer, thank you very much for the critical reading of the manuscript and the comments as well as suggestions to improve the quality of this manuscript.

We completely agree that food fraud largely impacts the economic gain. Therefore, a need for better discriminating methods to detect food fraud is urgent and regulations for official methods have to be updated. Even though the costs of purchasing equipment are high to bring reference laboratories up to date, publications underpinning the value of sensitive and high throughput laboratory equipment will facilitate the need for renovations.

Reviewer 2 Report

General comments

Manuscript is well-written and main goals are clear. However, this research is clearly skewed by the exclusion of liquid chromatography high resolution mass spectrometry analysis (LC-HRMS) to perform high output metabolite profiling of samples. Nowadays, LC-HRMS supports most of the analytical approaches to metabolite analysis with some exceptions such as the study of volatile compounds (GC-MS), unknown and isobaric species (NMR) and some targeted applications for routine analysis of waste, pesticides and food contaminants (tripleQ detection). Currently, LC-HRMS is by far the most capable technology in terms of sensitivity, flexibility (targeted/untargeted research), dynamic range, high throughput capacity and automation.

Length of the manuscript is excessive considering the analytical platforms of analysis used. Obviously, the weakest technology was the LC-UV/Vis and it was not necessary to include it in the research. In this line, best results were going to be provided by a standard GC-MS (single Q) and NMR detection. Taking into account the results presented by authors (almost the same) and the existing huge difference of price between both devices, we can consider that the GC-MS alternative is the most suitable for the majority of researchers. Then, same results would be achieved through the comparison of samples just using GC-MS analysis. In this line, it was obvious that the higher metabolites studied the higher discrimination will be achieved among samples and then, the targeted analysis of only 10 free amino acids (free AAs) by LC-UV/Vis would be irrelevant.

Abstract

The entire section is plenty of truisms.

-Discriminant capacity of targeted LC-UV/Vis approach (10 free AAs) is negligible compared to untargeted GC-MS and NMR approaches.

-The higher degree of hydrolyzation of the additives incorporated will increase the signal response.

-Discrimination capacity among samples will be increased by considering higher number of metabolites instead of just using 10 free AAs.

Introduction

As written, this section do not explain what is the relevance of this research to overcome problems in food fraud detection. Authors avoid to mention the outstanding capacity of LC-HRMS analysis for both exploratory and targeted approaches and the fact that nowadays it is the election of choice for hunting of metabolite/peptide biomarkers.

-Lines 51-55: entire paragraph is not correct. Free AAs and peptides are not detected by and HPLC, they are measured by detectors such as mass spectrometers that are coupled to an HPLC. Term “targeted “ at line 53 is not appropriate since when talking about LC-MS it can be strictly applied to TripleQ detection whereas exploratory devices such as qTOF, Orbitrap, Qtrap and ion-trap perform both targeted/untargeted qualitative/quantitative approaches. Statement about large datasets needed by LC-MS (reviewer considers that authors wanted to talk about MS analysis) approach to discriminate groups of samples can be applied to any other analytical strategy.

-Lines 63-64: this statement is not correct. Exploratory qualitative/quantitative metabolite analysis is mainly supported by MS detection. If we are talking just about quantitative research, main techniques are MS and NMR approaches.

Materials and methods

-Please define the UV/Vis detector. Reviewer envisages that it was used a photodiode array detector (PDA). Wavelengths used must be defined. There is no sense to use LC-UV/Vis analysis considering sensitivity and noise limitations, the few analytes assayed and the necessity of sample derivatization.

-Lines 183-184: normalization procedure of samples is very unclear.

-Line 186: why absolute quantitation of analytes in GC-MS analysis was done using external calibration instead of using the spiked internal standards (ISs)?.

 -Lines 221-234: why metabolites identified by NMR (86) were not listed in a supplementary table as done for GC-MS research (Table S4).

Results

Length of text is excessive just to demonstrate obvious achievements:

-LC-UV/Vis capacity is very limited and must be restricted to the analysis of only few analytes. Obviously, sensitivity shown by this approach was the lowest.

-The higher degree of protein hydrolyzation the higher signal response of free AAs.

-Discrimination capacity increases by increasing the number of analytes assayed. This can be only achieved by a preliminary untargeted screening of potential biomarkers samples followed by a targeted research of proposed candidates. This is the classic protocol for hunting of biomarkers.

Conclusions

Efficiency of discrimination was similar in GC-MS and NMR research. Authors concluded that NMR analysis must be the election of choice according to its higher output capacity and the unnecessary of sample derivatization. The first assumption is not correct since GC-MS alternative can handle bulky analysis thanks to automation (automated injection of hundreds of samples by autosamplers and automated on-the-fly data processing by dedicated software solutions). The second statement about sample derivatization is correct regarding GC-MS but again, the absence of LC-HRMS analysis in this research skewed final conclusions. LC-HRMS analysis is much more sensitive than NMR, can handle bulky sample analysis through automation and does not need sample derivatization. Then, reviewer encourages authors to carry out a similar research than this presented one but comparing LC-HRMS vs NMR. 

Considering results achieved and LC-HRMS technology currently available, significance of this research is very limited.

Author Response

Dear Reviewer, thank you very much for the critical reading of the manuscript and the comments as well as suggestions to improve the quality of this manuscript. We changed the manuscript as suggested and please find our answers attached.

Reviewer 3 Report

Introduction: you should well describe the link between aminoacids identification and food fraud. 
Therefore, you shoud clarify the relation between free aminoacids and REF, GPH, WPH, CPH, GTH, WTH, CTH and water. 

Line 19: why do you specify number 1 near N-HMR? You can write only H-NMR.
Line 31: Keywords are too enough. You can write about 5 or 6.
Line 50: There's a large space between "ingredients  and a further".
Line 51: You should well describe the mentioned methods.
Line 83: Materials and Methods are unclear; you should divide it into: 
             - Chemicals 
             - Instrument
             - Sampling
             - Calibration Procedures
             - Sample preparation
             - HPLC/UV-Vis - GC-MS and H-NMR Methods
             - Statistical Analysis
Line 101: D2O, what does it means? You can write chemical formula as well as other chemical compounds
Lines 131-135: You should well describe this part. It's unclear
Line 155: Why do you not use the same sample procedures for HPLC analysis?
Line 183: You should modify and control the number of supplementary tables. When you are writing for the first time a table, it will be numbered with 1. Table S4 is not present in Supplementary. You should control in whole paper that table number are correct.
Line 191: When you cite the paragraph with its number you should write: "2.4.2 paragraph". 
Line 206: Paragraph is unclear. 
Lines 245-247: In Material and Methos, you should well specify all the aminoacids that are analyzed in the manuscript. 
Lines 265-267: "Manuscript in preparation"? 
Figure 1 and 2: figures are unclear; you should adjust histograms scale to make them more clear.
Lines 288 and 315 (figure 1) and (figure 2) have to be entered in the text, not in the title. 
Lines 325 and 327: You should control capital letters after the colon.
Lines 338-341: Ypu should clarify this part in the introduction:
Lines 342-364: You should describe this part as well in the introduction instead of results and discussion.
Paragraph 3.2: you should describe it into a clear and short form. Why do you not insert PCA about HPLC data?
Lines 478-480: You should well describe figure in the text. Therefore, you should control dimensions of the text.

Author Response

(The authors gave the same response as above.)

Round 2

Reviewer 2 Report

Reviewer thanks the effort from authors to improve quality of the research. Current quality of the manuscript is better than originally however, this research is still skewed by the absence of LC-MS analyisis in the comparison.

As stated by authors, the only use of LC-UV/Vis analysis of free amino acids was not enough to achieve a reliable discrimination of samples. Then, the challenge was reduced to NMR and GC-MS approaches. Among these two techniques, GC-MS analysis is much preferred by Public Health institutions addressing quality assessment, security and all about traded foodstuffs. In this line, during the last years many GC-MS limitations were surpassed by LC-HRMS analysis, finding LC-HRMS detection the election of choice for most of the activities addressed by Public Health Institutions. In fact, the European Food Safety Authority (EFSA) strongly recommends the use of HRMS (mainly related to LC with fewer applications for GC) for almost all about food quality (safety, fraud, origin certification etc.) assessment in the EU area.

Evidently, affordability of LC-HRMS technology is still a limiting factor for many small labs but current prices are much more reasonable than five years ago. You can get interesting LC-HRMS systems for a better budget than NMR devices. In any case, interest of this research would be fulfilled by the inclusion of a LC-MS research even using a low-resolution, but modern, device such in the case of an exploratory ion-trap mass analyzer (as a counterpart of the GC-MS single-Q platform of analysis used by authors). Furthermore, analytical capacity of high-sensitive linear ion-traps for general exploratory analysis is significantly higher than NMR detection.

As previously mentioned, in its current state length of this research is excessive considering results achieved. Consideration for publication would be feasible if LC-MS analysis can be incorporated into the comparison. If not, reviewer recommends the publication of a lenght-reduced version of this study as a research note, rapid communication or similar but never as a full-research article.

Author Response

Dear Reviewer,

thank you very much for your critical discussion. Although the main focus of this study was the comparison of HPLC (targeted) and GC-MS as well as NMR (non-targeted) for the detection of food fraud. It is important to compare methods which are available and used by control authorities. 

Best regards!

Reviewer 3 Report

Article generally is well done. I suggest you to improve the English language throughout this article. You can find some corrections below:

Line 32: "ProHydrAdd"??

Line 165: You can modify this sentece adding "spectrophotometric analysis" and "wavelength between..."

Line 197: GC-MS solutions

Lines 198-201: You should review the English language. "Reduce for"

Line 205: "then"

Lines 212-213: write only once "Paragraph 2.4.2." without repeating it twice in the same sentence.

Line 256: i suggest you to write amino acids by their acronym, and maybe it will be suitable if you evidence them also in the introduction.

Author Response

Dear Reviewer,

thank you very much for you comments regarding english language.

Article generally is well done. I suggest you to improve the English language throughout this article. You can find some corrections below:

Line 165: You can modify this sentece adding "spectrophotometric analysis" and "wavelength between..."

As suggested, “spectrophotometric analysis” and “wavelengths between” was added (Line 168).

Line 197: GC-MS solutions

The software is called GCMSsolution, without an empty space.

Lines 198-201: You should review the English language. "Reduce for"

As suggested, we excluded “for” in the sentence.

Line 205: "then"

As suggested we changed corrected that word.

Lines 212-213: write only once "Paragraph 2.4.2." without repeating it twice in the same sentence.

As suggested, we excluded “according to paragraph 2.4.2” (line 215).

Line 256: i suggest you to write amino acids by their acronym, and maybe it will be suitable if you evidence them also in the introduction.

We never used the acronym of the amino acids in any part of the MS, therefore we think it might be to confusing if we acronym it here.